# Preoperative Cognitive Impairment and Postoperative Delirium Predict Decline in Activities of Daily Living after Cardiac Surgery—A Prospective, Observational Cohort Study

**DOI:** 10.3390/geriatrics5040069

**Published:** 2020-10-03

**Authors:** Ulf Guenther, Falk Hoffmann, Oliver Dewald, Ramy Malek, Kathrin Brimmers, Nils Theuerkauf, Christian Putensen, Julius Popp

**Affiliations:** 1University Clinic of Anaesthesiology, Intensive Care, Emergency Medicine and Pain Therapy, Klinikum Oldenburg, University Medicine Oldenburg, 26133 Oldenburg, Germany; 2Oldenburg Research Network Emergency- and Intensive Care Medicine (OFNI), Faculty VI—Medicine and Health Sciences, Carl von Ossietzky University, 26111 Oldenburg, Germany; oliver.dewald@uni-oldenburg.de; 3Department of Health Services Research, Division of Outpatient Care and Pharmacoepidemiology, Carl von Ossietzky University of Oldenburg, 26111 Oldenburg, Germany; falk.hoffmann@uni-oldenburg.de; 4University Clinic of Cardiac Surgery, Klinikum Oldenburg, University Medicine Oldenburg, 26133 Oldenburg, Germany; 5Department of Cardiology, Maria Hilf Hospital, 53474 Bad Neuenahr-Ahrweiler, Germany; rmalek@t-online.de; 6Clinic of Psychiatry, Psychotherapy and Psychosomatic Medicine, LVR Klinik Düren, 52353 Düren, Germany; kathrin.brimmers@gmx.de; 7Department of Anaesthesiology and Intensive Care Medicine, University Hospital Bonn, 26105 Bonn, Germany; nils.theuerkauf@ukb.uni-bonn.de (N.T.); christian.putensen@ukb.uni-bonn.de (C.P.); 8Old Age Psychiatry, Department of Psychiatry, Lausanne University Hospital, 1008 Prilly, Switzerland; Julius.Popp@chuv.ch; 9Centre for Gerontopsychiatric Medicine, Department of Geriatric Psychiatry, Hospital of Psychiatry Zürich, 8032 Zürich, Switzerland

**Keywords:** delirium, older people, cardiac, surgery, intensive care, outcomes, activity of daily living, cognition, cognitive

## Abstract

Cardiac surgery and subsequent treatment in the intensive care unit (ICU) has been shown to be associated with functional decline, especially in elderly patients. Due to the different assessment tools and assessment periods, it remains yet unclear what parameters determine unfavorable outcomes. This study sought to identify risk factors during the entire perioperative period and focused on the decline in activity of daily living (ADL) half a year after cardiac surgery. Follow-ups of 125 patients were available. It was found that in the majority of patients (60%), the mean ADL declined by 4.9 points (95% CI, −6.4 to −3.5; *p* < 0.000). In the “No decline” -group, the ADL rose by 3.3 points (2.0 to 4.6; *p* < 0.001). A multiple regression analysis revealed that preoperative cognitive impairment (MMSE ≤ 26; Exp(B) 2.862 (95%CI, 1.192–6.872); *p* = 0.019) and duration of postoperative delirium ≥ 2 days (Exp(B) 3.534 (1.094–11.411); *p* = 0.035) was independently associated with ADL decline half a year after the operation and ICU. Of note, preoperative ADL per se was neither associated with baseline cognitive function nor a risk factor for functional decline. We conclude that the preoperative assessment of cognitive function, rather than functional assessments, should be part of risk stratification when planning complex cardiosurgical procedures.

## 1. Introduction

Cardiac surgery and subsequent intensive care unit (ICU) treatment in patients at higher age have been shown to be associated with sustained cognitive impairment [1,2], being strongly correlated with the perceived quality of life [3]. Functional declines—apart from cognitive changes—upon cardiac surgery have also been described [4,5]. Another group found an initial functional decrease for four to six weeks, but no differences six months after cardiac surgery [6]. Others, though, have reported a functional improvement in up to 81% of their cases [7].

Most cardiac surgery patients “expect to return to normality“ [8]. The risk factors for functional decline have been reported to be—among others—impaired preoperative cognitive performance and postoperative delirium [2,4]. Delirium is particularly prevalent in cardiac surgery if extracorporeal circulation is required, as it provokes a non-infectious systemic inflammatory syndrome (SIRS) [9]. SIRS and sepsis have already been shown to be associated with postoperative delirium after cardiac surgery, which in turn can impair functional outcomes [4,5,10]. Another plausible risk factor appears to be impaired preoperative functional status. Frailty, assessed with a variety of tools, was shown to be associated with functional decline [11]. Others, however, have reported preoperative functional status not to be associated with postoperative outcomes [12]. 

These contradictory findings may be attributed to the variety of tools used to assess functional outcomes and the degrees thereof [11]. For this study, we focused on the activities of daily living (ADL) to establish the perioperative risk factors of functional decline half a year after cardiac surgery and intensive care treatment [10]. 

## 2. Materials and Methods

This is an a priori planned secondary analysis from a single-center prospective cohort study which enrolled patients scheduled for cardiac surgery in the University Hospital Bonn, Germany, with approval from the medical ethics committee of the University of Bonn, Chair Prof. Dr. Racké (No. 058/09, 28 May 2009), and registered in the German clinical trials registry (DRKS00021662). This study was conducted in accordance with the Declaration of Helsinki. During a 6-month period from July through December 2009, every consecutive patient scheduled for cardiac surgery was screened for eligibility, and followed up by telephone until June 2010. The exclusion criteria were age <50 years, inability for preoperative assessment of cognitive function (aphasia, psychiatric disease, insufficient knowledge of German language), participation in another study, as well as urgent and emergency surgery. 

After obtaining written informed consent, the patients’ demographics and clinical parameters—including age, gender, length of school and job education, weight, height, and gender—were recorded. The assessment of their past medical history included current medication, visual and hearing impairment, and nicotine and alcohol consumption. Nicotine consumption was classified as “never”, “stopped before at least two months”, and “ongoing”. Alcohol consumption was graded as “never”, “occasionally” (≤7 drinks per week), “moderate” (7–14 drinks per week), and “heavy” (>5–6 drinks per day.) A thorough assessment of organ dysfunctions, including cardiac, pulmonary, neurological, gastrointestinal, hematological, malignant, and connective tissue diseases was carried out, and the Charlson comorbidity index as well as the logistic EuroScore—derived from age, sex, and the number and seriousness of pre-existing comorbidities—were calculated [13,14].

ADL was assessed according to the “Alzheimer’s Disease Cooperative Study Group” (ACDS-ADL) pre-operatively and six months thereafter. ACDS-ADL includes the evaluation of instrumental activities of daily living [15]. ACDS-ADL is often preserved at the very early stages of cognitive impairment and—together with the Mini-Mental State Examination (MMSE)—may therefore be a tool to identify and monitor patients experiencing cognitive and functional decline over time [15,16]. Additionally, as ADCS-ADL is completed by a spouse or a close relative, it can be performed during a phone call and does not rely on the performance of the patient. Global cognitive performance and depressive symptoms were assessed with the MMSE and the Geriatric Depression Scale (GDS) [17]. Patients’ preoperative assessments, including the documentation of co-morbidities and concomitant medication, were carried out by medical Ph.D. students, who were repetitively re-trained by the psychiatrist of our research team. Patients’ blood samples were drawn for preoperative laboratory investigations, including hemoglobin, hematocrit, platelet count, white blood count, C-reactive protein, creatinine, sodium, and potassium. As carrying the ε4 allele of the apolipoproteine E gene (ApoE ε4) was previously proposed as a genetic risk factor for delirium in the elderly [18], patient’s DNA was isolated and the ApoE ε4 genotype determined as previously described [19].

The premedication consisted of midazolam (7.5 mg orally) on the evening before and on the morning of surgery. The induction of anesthesia included propofol, sufentanil, and pancuronium bromide. Anesthesia was maintained with continuous sufentanil and inhalative isoflurane. Dosages were left to the attending anesthetist’s discretion. Mild hypothermia (target temperature, 34 °C) was induced during cardio-pulmonary bypass. No antagonization of neuromuscular blockade was conducted. Transesophageal echocardiography was used to confirm the intraoperative deairing of the heart. All the patients were transferred sedated and mechanically ventilated to the ICU. The type of cardiac surgery, length of hypotension (mean arterial pressure <65 mmHg), length of catecholamine requirement (sum of epinephrine and norepinephrine dose >0.05 µg·kg^−1^·h^−1^), use of milrinone, lowest hemoglobin concentration during cardiopulmonary bypass (CBP), length of hypothermia (classified as T < 35 °C), lowest temperature, duration of cardiopulmonary bypass, duration of aortic cross clamping, use of intraaortic ballon counterpulsation (IABP), hypoglycemia (serum glucose < 60 mg/dL), and hyperglycemia (serum glucose > 150 mg/dL) were recorded.

Patients were sedated with continuous propofol and sufentanil in the ICU, and neuromuscular blockade was not part of the regular regimen. Additional pethidine was given in cases of shivering. Analgesia was regularly provided by piritramide as individual bolus (2–5 mg) intravenously and ibuprofen as a suppository three times a day. Beginning with the first day after cardiac surgery, patients were rated at least once per day shift and (if not sleeping) once per night for their state of vigilance with the Richmond Agitation Sedation Scale (RASS) [20] and their level of pain with the Behavioral Pain Scale (BPS) by the attending ICU physicians [21]. Delirium was assessed at the same time up to seven days during ICU stay with the Confusion Assessment Method for Intensive Care Unit (CAM-ICU) [22]. Patients with a RASS score of −4 or −5 were rated “unable to assess”. The “Simplified Acute Physiology Score” (SAPS II) to assess patients’ severity of medical condition were calculated from day charts and nurses’ notes on the day of admission to ICU [23]. Cardiovascular events including arrhythmias, cardiopulmonary resuscitation, need for mechanical ventilation, and continuous veno-venous hemofiltration, as well as signs of systemic inflammatory response syndrome (SIRS) [24] and any other clinical significant complications, such as re-thoracotomy and sternal infection, were daily recorded. The length of stay in ICU and in hospital as well as length of ventilation were retrieved from the hospital patient data managing system after discharge. Half a year after surgery, the patients and relatives were contacted by telephone to complete a 22-item telephone version of the MMSE (maximum score, 22 points) [25]. During the same call, the relatives were interviewed for patients’ ADL, if possible. The period of six months was chosen, because the patients were expected to have returned from hospital and rehabilitation by then. Additionally, we speculated that motivation to cooperate would be higher six months rather than a year after surgery.

Patients’ demographic data were grouped into whether they had postoperative ADL lower than preoperative ADL (“ADL decline”), or whether there was no change or a higher ADL than before surgery (“No decline”). Patients’ characteristics, including age, height, weight, EuroScore, laboratory results, SAPS II on the day of admission to ICU, duration of mechanical ventilation, length of stay in the ICU, and length of stay in hospital were calculated as a median with interquartile range (IQR) and compared with the Mann–Whitney U test. Categorical variables such as gender, comorbidities, type of surgery, and presence of ApoE ε4-allel were calculated as relative frequencies with percentages and compared with Fisher’s exact test or a Chi square-test, if appropriate. A *p* < 0.05 was considered statistically significant for all parameters. 

Factors associated with “ADL decline” in the univariate analysis (MMSE ≤ 26 points, duration of aortic clamping, length of delirium ≥2 days), as well as age, gender, and preoperative ADL were entered into a binary logistic backward regression analysis with “ADL decline” as the independent variable. A stepwise backward selection logistic regression analysis was used with an inclusion criteria of *p* ≤ 0.05 and an exclusion criteria of *p* ≥ 0.10. Statistical analyses was performed with IBM^®^ SPSS^®^ Statistics Version 26 (IBM Deutschland GmbH, Ehningen, Germany) and GraphPad Prism 5.0 (GraphPad Software, Inc.; San Diego, CA, USA).

## 3. Results

Four-hundred and one patients were screened, 162 of whom were not eligible for this study (Figure 1). Reasons for exclusion were age <50 years (*n* = 102), emergencies (*n* = 46), participation in another study (*n* = 5), and not German speaking (*n* = 9). Eighteen more patients declined or withdrew consent, leaving 221 patients for complete preoperative assessments. Out of these, three eventually did not receive surgery, and 12 more died in ICU or during the follow-up period. Twenty-eight patients did not have a spouse or close relative to complete the preoperative ADL questionnaires. For 37 patients, the postoperative ADL was not completed due to multiple reasons (e.g., signs of dementia, weakness, still in rehabilitation facility, stroke), 14 patients or their relatives declined to participate, and two patients were lost to follow-up, leaving 125 datasets for the analysis of decline in ADL. Of two patients, ADL could not documented before surgery, and two more patients’ relatives declined to complete the ADL questionnaire after surgery, but they all stated that the patients were “doing much worse” than before surgery. These four patients were grouped into “ADL decline”. They were excluded from the calculation of postoperative ADL, but were used otherwise to identify the risk factors of ADL decline.

Data from those patients of whom no follow-up information was obtained (N = 81) are available as online Appendix A. In brief, more woman than men were lost to follow-up, education was slightly but significantly shorter, and there was a significantly higher frequency of visual impairment and a lower frequency of cardiac insufficiency (LV-EF < 50%) in the “No follow-up” group. The lowest intraoperative temperature was lower in the “Follow-up” group (*p* = 0.049).

Table 1, Table 2 and Table 3 show a representative selection of preoperative patients’ characteristics and intra- and postoperative factors, divided into “ADL decline” and “No decline”. The majority of patients (60%; *n* = 75 of 125) with completed follow-up had a lower ADL 180 days after cardiac surgery and ICU. In those, the mean ADL declined by 4.9 points (95% CI, −6.4 to −3.5; *p* < 0.000, Wilcoxon signed rank). In the “No decline” group, the ADL inclined by 3.3 points (2.0 to 4.6; *p* < 0.001). 

Intraoperative patients’ parameters divided into “ADL-decline” and “No ADL-decline”. All the continuous data were compared with the Mann–Whitney U test and presented as medians with interquartile ranges (IQRs). Categorical variables are presented as the number and percentage of patients and were compared with two-sided Fisher’s exact test. ADL, activity of daily living. Vasopressors: sum of the highest dosage of epinephrine and norepinephrine.

Three variables differed between those with and without ADL decline: lower score of preoperative cognition (MMSE ≤ 26, *p* = 0.007), duration of aortic crossclamping (*p* = 0.046, Table 2), and length of delirium ≥2 days in ICU (*p* = 0.018, Table 3). We chose an MMSE ≤ 26 as the cut-off value because it had the highest odds ratio (3210; 95%CI, 1364–7550; *p* = 0.007) in the univariate analysis, as compared to higher and lower MMSE levels. For no other parameter, including age and preoperative ADL, was a statistically significant difference found even after computing them as dichotomous variables. 

When comparing patients with preoperative MMSE ≤ 26 and MMSE ≥ 27, no significant difference was found in the preoperative ADL (Figure 2a). The median preoperative ADL was 42 (IQR, 37–47) points in the “MMSE ≥ 27” group vs. 40 [35–45] points in the “MMSE ≤ 26” group (*p* = 0.084). The ADL remained stable in the “MMSE ≥ 27” group (postoperative ADL, 43 [36–46] points, *p* = 0.593), whereas there was a significant ADL decline in the “MMSE ≤ 26” group to 36 [28–40] points (*p* < 0.0001, Wilcoxon signed rank). 

Postoperative delirium did have an impact on ADL when lasting two days or longer (Table 3). Figure 2b shows that ADL declined from 42 [37–47] to 36 [33–43] points (*p* = 0.001, Wilcoxon signed rank test) in the group who had “Delirium ≥2 days”. ADL remained stable in the groups with no or only one day of delirium. Note that, again, the preoperative ADL did not differ between groups (ADL “No delirium” vs. “1 day of delirium”, *p* = 0.291; ADL “No delirium” vs. “≥2 days of delirium”, *p* = 0.911, Mann–Whitney test).

Factors associated with “ADL decline” in the univariate analysis (MMSE ≤ 26 points, duration of aortic clamping, length of delirium ≥2 days), as well as age, gender, and preoperative ADL were entered into a binary logistic backward regression analysis with “ADL decline” as the independent variable. Table 4 shows that both the preoperative “MMSE ≤ 26” and postoperative length of “delirium ≥ 2 days” were confirmed to be independently associated with ADL decline. Preoperative ADL was confirmed not to be a risk factor for “ADL-decline”.

Age, sex, MMSE ≤ 26, preoperative ADL, duration of aortic clamping, and length of delirium ≥2 days were included in a binary logistic regression analysis with the outcome variable, ADL decline. This model included *n* = 123 patients’ data sets. R^2^ = 0.094 (Cox and Snell); R^2^ = 0.126 (Nagelkerke); χ^2^(4) = 12.089; *p* = 0.002. CI, confidence interval; MMSE, Mini-Mental State Examination.

## 4. Discussion

This study was conducted to investigate the functional status before and after cardiac surgery and intensive care treatment, and to explore potentially modifiable risk factors of adverse outcomes. The data presented in this study revealed that the majority of patients (60%) experienced substantial functional decline half a year after elective cardiac surgery and intensive care treatment. The risk factors for “ADL decline” were preoperatively impaired global cognition (MMSE ≤ 26) and length of postoperative “delirium ≥2 days”. Preoperative functional status (ADL) per se was not a risk factor for “ADL decline”, and it was not associated with preoperative cognitive status. Factors reported by others to be associated with functional decline (ApoE ε4, higher creatinine) were not confirmed in this study [26].

### 4.1. Preoperative Cognitive Status and Risk of Functional Decline

Preoperative cognitive impairment has repeatedly been shown to be a risk factor of postoperative delirium and sustained cognitive impairment after cardiac surgery [2,10,27,28]. As a measure of cognitive performance, the MMSE was used here since it had become standard in this field of research. MMSE alone as a single tool, however, is not sufficient for the diagnosis of cognitive impairment. The purpose here was to identify single factors to identify patients at risk of functional decline in the clinical setting. Focusing on functional decline, another prospective study involving 356 cardiosurgical patients found “cognitive functioning at baseline” to be independently associated with functional decline 1 year after cardiac surgery [26]. In line with these reports, our data clearly demonstrate, that preoperative cognitive status is not only a risk factor for postoperative delirium, but also an independent risk factor for subsequent functional decline.

### 4.2. Postoperative Delirium and Risk of Functional Decline

Delirium has been previously identified as an independent predictor of postoperative cognitive impairment up to one year after cardiac surgery [2,4], while others found a return to preoperative baseline at 12 months [29]. They had, however, a rather low rate of delirium and did not assess functional status.

In a prospective study involving 300 elective cardiosurgical patients, delirium was not only associated with increased mortality, frequent hospital readmissions, and reduced quality of life, but also with functional impairment, such as reduced independency in ADL and mobility [5]. In non-cardiac patients (*n* = 566), delirium was risk factor for “clinically meaningful” functional decline 18 months after surgery [30].

Interestingly, they also demonstrated that the duration of delirium did have an impact: one month after surgery, each additional day of delirium increased the risk of functional decline significantly [4]. That relationship, however, could not be established at one year after the operation [4]. The severity of delirium might also play a role; in a study comparing transcatheter and surgical aortic valve replacement, patients with more severe delirium had worse functional status trajectories [9]. Our present work underlines the importance of the duration of delirium not only for mortality, but also for functional outcomes.

### 4.3. Preoperative Functional Status

Only a few studies have evaluated functional status preoperatively and followed patients up postoperatively for more than four to six weeks. The comparison of studies is difficult due to a variety of assessment tools used at different postoperative time points.

In a study involving 62 patients after cardiac surgery and using a count of nine pre- and postoperative ADLs, seven of which were instrumental ADLs (IADL), pre-operative ADL impairment was associated with ADL decline 1 month postoperatively, but was not discriminative of functional outcomes six months after surgery [6]. Others, likewise, found preoperative IADLs not to be predictive of functional outcomes one year after surgery [26,31]. The degree of functional impairment seems to matter; in a study in 428 elective cardiosurgical patients using groups of physical activity (PA), the lowest PA (“sedentary level”) was an independent risk factor for no functional improvement [32]. Their results also showed that patients with an “active level” were unlikely to benefit from cardiac surgery [32]. In our work, we also tried to group patients according to their preoperative ADL, but no cut-off value could be determined.

The temporal course of postoperative assessments should be paid attention to when comparing studies on functional long-term outcomes. In a study on 372 non-cardiosurgical patients, the maximum functional declines occurred 1 week postoperatively [33]. The mean recovery times depended on the chosen outcome parameter; MMSE had a recovery time of three weeks, and timed walk required six weeks. ADL and functional reach took 3 months and IADL took 6 months to recover from. The mean grip strength did not return to the preoperative status by six months [33]. Of note, Min et al. found a functional impairment only at 4–6 weeks, but not half a year after surgery, whereas our research found a significant difference at the same time point [6]. Conversely to the findings of Min et al., a lower ADL six weeks after surgery was reported to predict ADL decline six months after surgery [31]. Rudolph et al. found substantial functional deficits even a year later in a cohort of 191 patients using an array of assessment tools [4]. These differences may be attributable to different sample size and cohort characteristics, but are also certainly attributable to the different assessment tool that have been employed. 

### 4.4. Limitations

Some major limitations should be addressed: Firstly, we investigated only elective cardiosurgical patients of higher age. This led to a rather homogenous patient group. Therefore, age and creatinine were not found to be risk factors of functional decline, in contrast to other reports [26]. The risk factors of functional decline presented in this work must therefore be interpreted with caution and cannot be easily extrapolated to other patient populations, such as, for instance, medical or emergency patients. Secondly, we focused on patients’ ICU stay. There was a standard for preventing and managing delirium established in our institution, consisting of early mobility and exercise, restricting drug treatment to haloperidol and clonidine, and opioids and non-steroidal anti-inflammatory drugs used for pain treatment. However, we could not control for this, nor can we reliably exclude other psychotropic medications or more opioids during their stay in the general ward or in the subsequent rehabilitation units. Thirdly, more than 1/3 of patients were lost to follow-up, including many with missing assessments of telephone ADL. Since that group showed no differences in preoperative parameters compared to the “No-decline” group, no major bias should be expected to be caused by this rather large proportion of drop-out patients.

Fourth, the postoperative assessment of the MMSE was performed by a telephone version with a maximum of only 22 points. It skips those tasks that require a face-to-face situation, such as drawing the interlocking pentagons, etc. We found a significantly lower telephone MMSE in the “ADL-decline” cohort. Though the 22-item telephone MMSE was shown to strongly correlate with the full MMSE [34], we did not attempt to compare it with the pre-operative MMSE, since we felt that a telephone MMSE performed at home half a year later should not be compared to a face-to-face situation in hospital. For instance, diminished hearing was shown to be associated with lower scores on the telephone MMSE [25]. As this was the case for all the participants, the lower telephone MMSE in the “ADL-decline” group might be taken as a hint for a cognitive decline, but no further conclusions should be drawn. 

Finally, we used only ADL to assess functional status. In non-surgical patient groups, functional status and cognition are closely associated [35]. This may not be the case in surgical patients, as their underlying medical conditions, requiring cardiac surgery, may per se be associated with ADL impairment [36]. Accordingly, even a significant increase in IADL six months after cardiac surgery has been reported in a series of 551 patients followed up for one year using a large battery of assessment tools [37]. Unfortunately, this increase in IADL did not lead to a higher “quality of life” (QoL), as it was found to be diminished by postoperative cognitive decline [36]. As we did not assess QoL specifically, no conclusion can be drawn as to whether a change in ADL affected patients’ QoL. 

## 5. Conclusions

The majority of patients had a substantial decline in ADL after cardiosurgical ICU. A lower preoperative cognitive score and ICU delirium ≥2 days were independent risk factors for ADL decline. Preoperative cognition and baseline functional status were statistically not associated. Hence, additional work should be directed towards tool development for the reliable detection of risk factors for post-operative functional decline. Ideally, these should include brief measures of cognition, functional status, and QoL, and it should be possible to perform them during follow-ups on the telephone. They would be relevant for the future development of interventions to prevent functional decline in older people undergoing cardiac surgery.

## Figures and Tables

**Figure 1 geriatrics-05-00069-f001:**
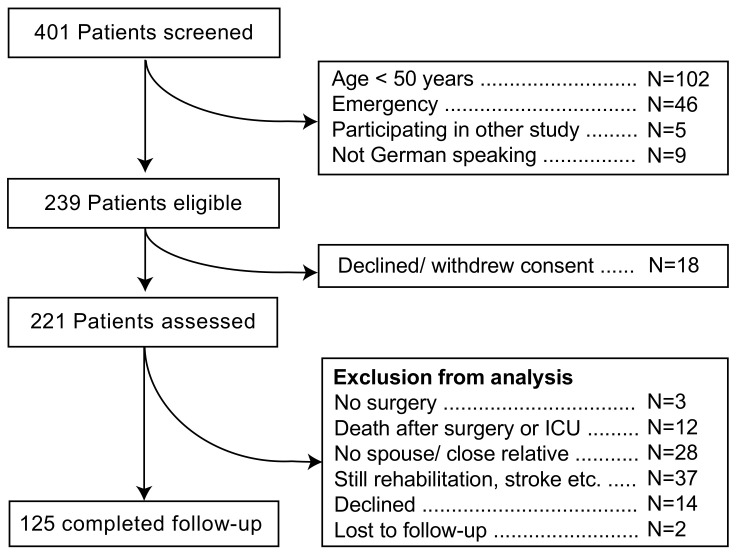
Patients’ recruitment.

**Figure 2 geriatrics-05-00069-f002:**
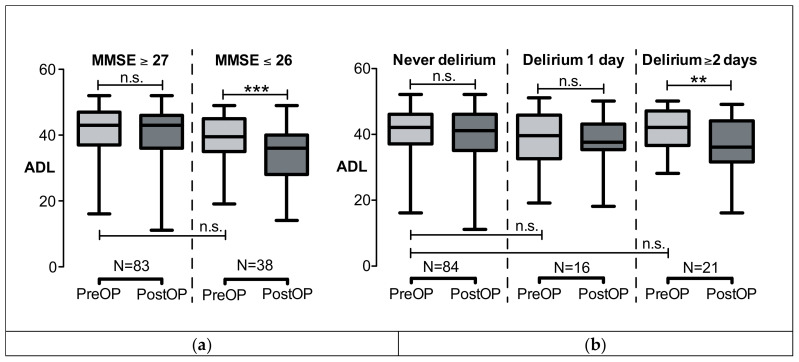
ADL before (preOP) and half a year after cardiac surgery and intensive care (postOP). ADL did not pre-operatively significantly differ between groups. (**a**) ADL was divided into MMSE ≥ 27 or MMSE ≤ 26 before surgery. ADL significantly declined only in the MMSE ≤ 26 group, ADL remained stable in the MMSE ≥ 27 group. (**b**) ADL was divided into whether patients had no delirium or 1 day or ≥2 days of delirium. ADL significantly declined if the delirium lasted ≥ 2 days. Data are shown as boxplots (median; minimum, maximum; 25% and 75% quartile). *** *p* < 0.001, ** *p* < 0.005; n.s., not significant. ADL, activity of daily living; MMSE, mini mental state examination.

**Table 1 geriatrics-05-00069-t001:** Patients’ characteristics.

	ADL Decline, *n* = 75		No ADL Decline, *n* = 50		*p*
	Median	IQR	Median	IQR	
Age (years)	73	[64–77]	71	[62–77]	0.425
Body mass index	27	[25–30]	26	[25–29]	0.704
ADL (points)	42 ^#^	[38–47]	42 ^#^	[35–45]	0.327
GDS (points)	2	[1–3]	1	[1–3]	0.615
Length of education (years)	11	[10–14]	11	[11–13]	0.855
Total protein (g/L)	45 *	[40–48]	46 *	[41–49]	0.340
Charlson (points)	2	[1–3]	2	[1–3]	0.054
EuroScore (points)	6	[4–7]	6	[4–8]	0.646
Hemoglobin (g/dL)	13.8	[12.6–14.6]	13.6	[12.5–14.6]	0.690
Platelet count (G/L)	221	[182–264]	231	[192–283]	0.316
WBC (G/L)	7.4	[6.0–8.8]	6.8	[5.7–8.0]	0.168
CRP (mmol/L)	3.0	[1.5–7.0]	2.2	[1.0–5.9]	0.098
Creatinine (mg/dL)	1.2	[1.0–1.4]	1.2	[0.9–1.3]	0.059
	N	%	N	%	*p*
Male/female ^‡^	59/16	79/21%	35/15	70/30%	0.296
MMSE ≤ 26	31	41%	9	18%	0.007
Age 50–59 years	12	16%	8	16%	1.000
Visual impairment	75	100%	48	96%	0.158
Hearing impairment	15	20%	12	24%	0.660
Atherosclerosis	6	8%	7	14%	0.372
Other than isolated CABG	41	55%	27	54%	1.000
Ascending aortic repair	4	5%	1	2%	0.647
Myocardial infarct	19	25%	14	28%	0.836
LV-EF < 50%	61	81%	38	76%	0.506
NYHA IV	2	3%	0	0%	0.516
Chronic atrial fibrillation	11	15%	8	16%	1.000
Chron. pulmonary disease	12	16%	3	6%	0.158
Nicotine					
Never	41	55%	28	56%	
Stopped >2 months	30	40%	15	30%	^‡^ 0.182
Ongoing	4	5%	7	14%	
Alcohol					
Never	19	25%	17	34%	
Occasionally	39	52%	23	46%	^‡^ 0.651
Moderate	14	19%	7	14%	
Heavy	3	4%	3	6%	
Chron. renal insufficiency	10	13%	4	8%	0.402
Diabetes mellitus 2	19	25%	7	14%	0.177
History of malignancy	9	12%	6	12%	1.000
History of stroke	2	3%	0	0%	0.516
APOEε4 carrier	12	19%	16	34%	0.121

Preoperative patients’ demographics and risk factors, divided into “ADL-decline” vs. “No ADL-decline”. Continuous data were compared with the Mann–Whitney U test and presented as median with interquartile range (IQR). Dichotomous data are presented as the number and percentage of patients and were compared with the two-sided Fisher’s exact test or Pearson’s Chi-Square test, the latter indicated by ^‡^. ADL, activity of daily living; CRP, C-reactive protein; GDS, geriatric depression scale; MMSE, mini-mental state examination; WBC, white blood cell count; LV-EF, left ventricle ejection fraction; NYHA, classification of heart failure according to the New York Heart Association; APOE ε4, carrier of ε4 allele of the apolipoproteine E gene. * Total protein: preoperatively, *n* = 60; postoperatively, *n* = 42; ^#^ Preoperative ADL, *n* = 73; postoperative ADL, *n* = 71.

**Table 2 geriatrics-05-00069-t002:** Intraoperative parameters.

	ADL Decline			No ADL Decline			*p*
	Median	IQR	N	Median	IQR	N	
Duration cardiac bypass (min)	127	[112–150]	70	118	[94–143]	47	0.085
Aortic clamping time (min)	81	[66–113]	70	70	[54–102]	47	0.046
Duration ≤ 65 mmHg (min)	120	[68–163]	68	115	[80–160]	47	0.838
Deepest temperature	34.0	[33.8–34.3]	68	34.1	[33.8–34.4]	47	0.218
	N	%		N	%		*p*
Vasopressors >0.05 µg·kg^−1^·h^−1^	61	88%	69	40	85%	47	0.779
Milrinone (use of)	11	16%	69	5	11%	47	0.585

**Table 3 geriatrics-05-00069-t003:** Postoperative parameters.

	ADL Decline			No ADL Decline			*p*
	Median	IQR	*n*	Median	IQR	*n*	
SAPS II on admission to ICU	32	[27–37]	75	30	[25–34]	50	0.155
Packed red cells (units)	4	[2–6]	75	3	[2–4]	50	0.529
Ventilator days	1	[2–3]	75	1	[1–2]	50	0.066
Length of stay in ICU (days)	2	[1–5]	75	2	[1–4]	50	0.204
Stay in-hospital (days)	15	[10–21]	75	13	[10–21]	50	0.512
ADL/180 days	37	[32–44]	73	44	[38–48]	50	0.000
MMSE/180 days, telephone	19	[16–21]	70	21	[20–22]	50	0.000
Delirium, length of	*n*	%		*n*	%		*p*
0 days	46	61.3%		40	80%		
1 day	10	13.3%		6	12%		0.04 ^‡^
≥2 days	19	25.3%		4	8%		
Delirium, subtype							
hypoactive	16	55%		5	50%		
hyperactive	6	21%		4	40%		0.398 ^‡^
mixed	7	24%		1	10%		

Postoperative patients’ parameters were divided into “ADL-decline” and “No ADL-decline”, respectively. Continuous data were compared with the Mann–Whitney U test and presented as medians with interquartile ranges (IQRs). Categorical variables are presented as the number and percentage of patients and were compared with two-sided Fisher’s exact test or Pearson Chi-Square test (indicated by ^‡^). ADL, activity of daily living; SAPS, Simplified Acute Physiology Score.

**Table 4 geriatrics-05-00069-t004:** Multiple regression analysis.

Variable	Regression Coefficient	Exp(B)	Exp(B) 95% CI	*p*
MMSE ≤ 26	1.051	2.862	1.192–6.872	0.019
Delirium, length ≥2 days	1.262	3.534	1.094–11.411	0.035
Constant	−0.105			0.651

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
