# Peer review of "Preoperative Cognitive Impairment and Postoperative Delirium Predict Decline in Activities of Daily Living after Cardiac Surgery—A Prospective, Observational Cohort Study"

_geriatrics, 2020, doi:10.3390/geriatrics5040069_

Round 1

Reviewer 1 Report

Thank you very much for the opportunity to read the results of the study on cognitive dysfunction in patients undergoing cardiac surgery. Undoubtedly, the assessment of cognitive functions is difficult, time-consuming and requires a lot of commitment. Therefore, I thank the authors for the effort they put into carrying out this research.

Below are my comments.

  1. Based on the pathophysiological knowledge, it can be concluded that hypotension followed by cerebral hypoperfusion may have an impact on long-term treatment outcomes. According to the research methodology, the authors collected data on hemodynamic parameters and pharmacological support of the circulatory system in the perioperative period. Were there any differences in hemodynamic stability between the "ADL decline" and "No ADL decline" groups? Additionally, blood transfusion is an important risk factor for cognitive decline / delirium. Can the authors verify the impact of this factor on postoperative cognition and ADL?
  2. Did the authors assess nutritional parameters?
  3. Some researchers suggest that the cut-off point for the diagnosis of disorders in MMSE should be adjusted to the patient's education (e.g. 24 points in a patient with higher education and 22 points in a patient with primary education). Please consider this aspect when assessing a patient's preoperative condition. Please consider performing an AUROC analysis to determine the MMSE cut-off point that best predicted "ADL decline" after 6 months to ensure that the conversion of the quantitative variable (MMSE score) to a dichotomous variable (MMSE <26 or> 26) did not result in the loss of valuable data.
  4. Suggest adding a figure showing the recruitment process of the study group (flow-chart), additionally, please verify the values in the "Results" section (Line 147-156):

238-18=220

220-6=214

214-28=186

186-37=149

149-14=135

135-2-9=124 patients

If there was an error, please verify the correctness of further statistical calculations in the manuscript.

  1. Did the authors identify subtypes of delirium in patients with postoperative delirium? Hypoactive delirium is the most common form of delirium in elderly patients. At the same time, it is the most difficult to diagnose and is often left untreated, in contrast to agitated patients who receive neuroleptics more often. This element could potentially affect your results. Please refer to this in the limitations. I think that should also be clarified in the text who had to examine the CAM-ICU algorithm (doctor? nurse?). I suggest clarifying in the manuscript whether pharmacological or non-pharmacological methods of delirium prophylaxis have been routinely used. If so, what methods? Supplementing this information will allow the publication to be included in future meta-analyzes.
  2. Some patients have been given pethidine, which increases the risk of delirium and worsens long-term treatment outcomes, including ADL. I suggest you include this fact in the Limitations section. In addition, it is currently suggested that midazolam should be discontinued, especially in elderly patients as it increases the risk of delirium. Clinical practice in 2009-2010 may be inconsistent with the current knowledge, therefore in the future I suggest publishing data earlier than 10 years after the end of the study.
  3. Were patients monitored during the first 6 months after the end of the study? I am interested in whether patients were routinely provided with a rehabilitation program until the ADCS-ADL test was retested. If not, there is a risk that some patients receive better or worse rehabilitation. Of course, the authors probably had no influence on it, but it is worth considering it as a limitation of inference.

Minor comments:

  1. Line 32 (abstract) - I suggest you delete "wilcoxon signed rank" - irrelevant in the abstract
  2. Line 142-143: „inclusion criteria of P≤05 and an exclusion criteria of P≥0.1” – error?
  3. Line 183, Table 2. “Delirium, length of…” column “ADL decline” – 61%+13%+25%=99% - error?
  4. Table 1 and Table 2. “N” instead “N=” in heading.
  5. Line 174, column “No decline/IQR”, line 5 (charlson points) - missing ")"
  6. Line 270 - please use the abbreviation MMSE
  7. Line 305 – please use the abbreviation “QoL” instead `quality of life`.
  8. Line 360 – I suggest detailing: “preoperative cognitive score and ICU-delirium lasting ≥2 days”

Author Response

Dear Reviewer,

thank your for your diligence. Your comments substantially improved our work ! Please find below our replies to your comments. In the re-uploaded manuscript, all changes were highlighted in yellow color. Our comments here were highlighted in orange.

1.Based on the pathophysiological knowledge, it can be concluded that hypotension followed by cerebral hypoperfusion may have an impact on long-term treatment outcomes. According to the research methodology, the authors collected data on hemodynamic parameters and pharmacological support of the circulatory system in the perioperative period. Were there any differences in hemodynamic stability between the "ADL decline" and "No ADL decline" groups?

An additional table was added (now Table 2) to give more information on haemodynamic parameters. Except for “duration of aortic clamping”, there were no significant differences between groups. Including “duration of aortic clamping” did not change the parameters of the multiple regression analysis.

Additionally, blood transfusion is an important risk factor for cognitive decline / delirium. Can the authors verify the impact of this factor on postoperative cognition and ADL?

The number of packed red cells was included in the above mentioned Table 3. There were no significant differences between groups.

2.Did the authors assess nutritional parameters?

Not really. We do have preoperative total protein (TP) in most patients. This information was included in Table 1. There was no significant difference between groups. We did not assess other features of frailty, such as gait speed, as it this was deemed too dangerous in cardiac surgery patients.

3 Some researchers suggest that the cut-off point for the diagnosis of disorders in MMSE should be adjusted to the patient's education (e.g. 24 points in a patient with higher education and 22 points in a patient with primary education). Please consider this aspect when assessing a patient's preoperative condition. Please consider performing an AUROC analysis to determine the MMSE cut-off point that best predicted "ADL decline" after 6 months to ensure that the conversion of the quantitative variable (MMSE score) to a dichotomous variable (MMSE <26 or> 26) did not result in the loss of valuable data.

This is true – more detailed information should be given here. Please find below the AUROC-analysis and the odd’s ratios (MMSE and ADL-decline):

MMSE     AUROC  (Std. error)   Asy. Sign.  OR           (95%CI)                  P

≤25        0.577     (0.051)          0.147          3.054       (1.058-8.817)         0.38

≤26        0.597     (0.051)          0.068          3.071       (1.209-7.800)         0.12

≤27        0.617     (0.050)          0.027          3.210       (1.364-7.550)         0.007

≤28        0.613     (0.051)          0.032          2.641       (1.228-5.678)         0.016

≤29        0.560     (0.053)          0.025          1.625       (0.789-3.345)         0.203

The results part now reads (line 214):

“We chose a MMSE ≤26 as the cut-off value for analysis, because it had the highest odds ratio (32.210 (95%CI, 1.364-7.550), P=0.007), compared to higher and lower MMSE-levels.”

Concerning education:

Thank you for this important comment. Subject’s education level certainly modifies cognitive performance and MMSE score. Also, diagnosis of cognitive impairment based on MMSE only is limited even if one considers education level. Rather than considering diagnosis based on complex evaluation and expert opinion as it is required for cognitive impairment or dementia, our purpose here was to identify single factors or clinical measures that predict functional decline.

This point is now addressed in the discussion section (lines 257 -261):

“As measure of cognitive performance, the MMSE was used, since it had become a standard in this field of research. MMSE alone, as a single tool, is not sufficient for the diagnosis of cognitive impairment. The purpose here was to identify single factors to identify patients at risk of functional decline in the clinical setting.”

We also added “length of education” (years) to Table 1. There was no significant difference between groups.

4.Suggest adding a figure showing the recruitment process of the study group (flow-chart), additionally, please verify the values in the "Results" section (Line 147-156):

238-18=220

220-6=214

214-28=186

186-37=149

149-14=135

135-2-9=124 patients

If there was an error, please verify the correctness of further statistical calculations in the manuscript.

Well noticed ! This was a typing error. All calculations and statistics were performed with the correct numbers. As suggested, we included a figure (Figure 1) to clarify the recruitment progress.

5.Did the authors identify subtypes of delirium in patients with postoperative delirium? Hypoactive delirium is the most common form of delirium in elderly patients. At the same time, it is the most difficult to diagnose and is often left untreated, in contrast to agitated patients who receive neuroleptics more often. This element could potentially affect your results. Please refer to this in the limitations.

This information about delirium subtypes is now given in Table 3. There were no significant differences between subtypes of delirium. It is true – hypoactive delirium is often missed if it is not properly looked after by using a dedicated delirium monitoring tool. The missing of hypoactive delirium should therefore be prevented by assessing patients for delirium on a regular basis, as it was done for this study by using the CAM-ICU. So we think, this a strength of our study, rather than a limitation.

I think that should also be clarified in the text who had to examine the CAM-ICU algorithm (doctor? nurse?).

Concerning the assessors, the ‘methods’-section now reads (line 92 – 95):

“Patients’ preoperative assessments, including documentation of co-morbidities and concomitant medication were carried out by medical Ph.D.-students, who were repetitively re-instructed by the psychiatrist of our research team.”

Concerning the postoperative assessments (line 116 - 119):

“Beginning with the first day after cardiac surgery, patients were rated at least once per day-shift and (if not sleeping) once per night for their state of vigilance with the Richmond Agitation Sedation Scale (RASS) (20) and level of pain with the Behavioral Pain Scale (BPS) by the attending physician (21). Delirium was assessed at the same time up to seven days during ICU stay with the Confusion Assessment Method for Intensive Care Unit (CAM-ICU) (22).“

I suggest clarifying in the manuscript whether pharmacological or non-pharmacological methods of delirium prophylaxis have been routinely used. If so, what methods? Supplementing this information will allow the publication to be included in future meta-analyzes.

True. The following was added to the ‘limitations’-section (line 316 – 321):

“There was a standard for preventing and managing delirium established in our instituion, consisting of early mobility and exercise, restricting drug treatment to haloperidol and clonidine, and opioids and non-steroidal anti-inflammatory drugs for pain treatment. But we did not control for this nor can we reliably exclude other psychotropic medication or more opioids during their stay in the general ward or in the subsequent rehabilitation unit.”

6.Some patients have been given pethidine, which increases the risk of delirium and worsens long-term treatment outcomes, including ADL. I suggest you include this fact in the Limitations section. In addition, it is currently suggested that midazolam should be discontinued, especially in elderly patients as it increases the risk of delirium. Clinical practice in 2009-2010 may be inconsistent with the current knowledge, therefore in the future I suggest publishing data earlier than 10 years after the end of the study.

As mentioned above, due to different periods of post-ICU stay in hospital, we cannot rule out any opioid analgesics, that were presumably given in the normal ward, or other drugs or procedures that might interfere with outcomes, such as – for instance – immobilization.

As mentioned above, we included a statement to the ‘limitations’-section (line 316 – 321).

7.Were patients monitored during the first 6 months after the end of the study? I am interested in whether patients were routinely provided with a rehabilitation program until the ADCS-ADL test was retested. If not, there is a risk that some patients receive better or worse rehabilitation. Of course, the authors probably had no influence on it, but it is worth considering it as a limitation of inference.

All patients subsequently went to some kind of rehabilitation program, either cardiac or neurological, and at different lengths. We did not control for the effects of different rehabilitation programs and different durations of these. As mentioned above, this was commented on in the ‘limitations’-section (line 316 – 321).

Minor comments:

1.Line 32 (abstract) - I suggest you delete "wilcoxon signed rank" - irrelevant in the abstract

In the abstract, "wilcoxon signed rank" was deleted as suggested.

2.Line 142-143: „inclusion criteria of P≤05 and an exclusion criteria of P≥0.1” – error?

The line should read: „inclusion criteria of P ≤ 0.05 and an exclusion criteria of P ≥ 0.10”.

3.Line 183, Table 2. “Delirium, length of…” column “ADL decline” – 61%+13%+25%=99% - error?

This is a rounding error, that’s true. This was corrected, the first decimal place is now given, which still does not prevent a small rounding error of 0.1 %.

4.Table 1 and Table 2. “N” instead “N=” in heading.

This was corrected.

5.Line 174, column “No decline/IQR”, line 5 (charlson points) - missing ")"

The bracket was inserted.

6.Line 270 - please use the abbreviation MMSE.

MMSE was inserted.

7.Line 305 – please use the abbreviation “QoL” instead `quality of life`.

This abbreviation is introduced only at this place, and is being used thereafter.

8.Line 360 – I suggest detailing: “preoperative cognitive score and ICU-delirium lasting ≥2 days”

Alright – this was done. Thank you !

Again - thank you your meticulous work !

Sincerely,

Ulf Guenther

Reviewer 2 Report

Thank you for the opportunity to review your work.

Guenther and colleagues performed a secondary analysis of a prospective cohort study to investigate perioperative risk factors of functional decline (focusing on activities of daily living (ADL)) 6 months after cardiac surgery and intensive care unit stay.

The study is important, as an emerging body of literature suggests that cardiac surgery and subsequent intensive care treatment in patients of higher age are associated with sustained cognitive impairment as well as functional decline, being strongly correlated with perceived quality of life. The detection of risk factors for post-operative unfavorable outcomes can be challenging, so the results are certainly valuable.

I have outlined several questions/comments below:

Introduction section

  1. The authors focus on activities of daily living (ADL) to establish perioperative risk factors of functional decline half a year after cardiac surgery and intensive care treatment. I wonder why the focus is on ADL and not on frailty, which is often the case in other studies and could have contributed to better comparability? Because on page 2 in line 56 the authors briefly discuss frailty. Furthermore, it is not quite clear to me from the introduction why the period of 6 months was chosen as the observation period. Is there a reason for this? Because the authors themselves mention in their abstract (pg.1, line 27-28) that statements regarding unfavorable outcomes are difficult to make because of the different assessment periods used in other studies.

Materials and Methods section

  1. From a statistical perspective, it remains unclear to me why no case number planning or power analysis was conducted for this study. Furthermore, it is a secondary analysis, for what were the corresponding data from 2009 and 2010 already used and why was a new recruitment not started again for the current study, since on the one hand the data were already used as well as are 10 years old and on the other hand it is evident in the results section (pg. 3, line 147-164) that many missing values exist in the current sample.
  2. Regarding the inclusion and exclusion criteria: Only patients over 50 years of age are included (pg. 2, line 70). However, it is not quite clear to me what the age range in this study was (age 62-77?) or on what age range the definition of elderly patients (pg. 1, line 27) in this study is based on. Furthermore, I ask myself whether all psychiatric diseases were generally excluded?
  3. For me it is still unclear, how, by what means and who obtained the demographic and clinical data (pg. 2, line 73-74) of the respective patients? Or were they taken all from the patient's medical record?
  4. Furthermore, how long did the examination/assessment last for the patients with regard to the study relevant parameters? (pg. 2, line 73-92)
  5. The authors made the following classification for nicotine consumption: "never", "stopped before at least two months", "ongoing". Why is there no "occasionally" classification, as for alcohol consumption? Furthermore, it is not clear to me whether the alcohol consumption is related to the type of drink (beer or wine etc.). Or has the definition of a standard drink (in the United States, one "standard" drink (or one alcoholic drink equivalent) contains roughly .6 fl oz or 14 grams of pure alcohol) been considered for this? Because it is not only the quantity alone that could make a difference.
  6. The authors decided to use the ACDS-ADL for the measurement of the functional decline. It is not clear to me why the authors chose this particular assessment tool. Because the ACDS-ADL normally evaluates the competence of patients with Alzheimer's disease (AD) in basic and instrumental activities of daily life (ADLs). This study did not exclusively include patients with AD.
  7. The decision for ACDS-ADL is referred to in one sentence (pg. 2, line 85-87), but a more detailed explanation or specification of the quality criteria of the assessment tool would be helpful.
  8. The authors make a classification of the patients in who had or did not have functional decline based on the ADCS-ADL (pg.2, line 91-92). This raises the question whether there was a corresponding cut-off for this
  9. After 6 months, the authors contacted the patients' relatives to perform the telephone version of the MMSE (pg. 3, line 127-129). How it was handled, since the maximum score of the telephone version was 22 points compared to the face to face version, where the maximum score was 30 points? Furthermore, how it was dealt with when no relatives could be reached? A more detailed explanation would be useful here.

Results section

  1. When the inclusion and exclusion criteria were first mentioned (pg. 2, line 70-72), the authors did not mention that participation in another study was an exclusion criterion, but they did mention it here (pg. 4, line 148). Please clarify.
  2. The authors were able to determine from their data that more men than women were lost in follow-up (pg. 4, line 162). Were there reasons for this? This would be interesting to know for the conduct of future studies, so that this circumstance could be counteracted in advance.
  3. For the formatting of Table 1 (pg. 4, line 174 as well as pg. 5) and Figure 1 a) and b) (pg. 6, line 200-208) see comment "Finally".

Discussion section

  1. The authors should, as usual in a discussion, briefly mention in a few sentences the aim of the study before summarizing the results and discuss these (pg. 6, line 221).
  2. In the section limitations on page 8, line 299-306, the authors have inserted the exact same text passage as line 293-299 a second time. In addition, an error has crept into this repeated text passage in line 300 after reference 33, in the way that the journal name and the year appear in the text.

Conclusion section

  1. The authors conclude that additional work should be directed towards tool development for reliable detection of risk factors for post-operative functional decline (pg. 8, line 311-313). This is important in view of the large number of different assessment tools. It would be nice to add recommendations or ideas for the development of such a reliable tool, which have the authors drawn from their own experience or could they suggest.

Miscellaneous

  1. Finally, the article is inconsistently formatted. The formatting would have to be revised again to make everything consistent. E.g. Introduction (on line 43) should be the headline on the page where the corresponding text passage is located (the same on line 146 concerning results).
  2. Regarding table 1 and figure 1: The percentage signs (%) in the table should be formatted without spaces behind the numbers and uniformly positioned one below the other (Table 1 on line 174). In figure 1, the a) and b) should be on the same level (pg. 6, line 200). In addition, on page 6 in line 207 lacks the corresponding explanation of the **.

Minor points:

- Page 3, line 136 as well as line 137: strange characters (they are not displayed correctly for Apo 4 allele and Chi. The same applies to page 7, line 226).

- Page 6, line 225: One blank too many before factors.

- Page 8, line 296: Accordingly, even a significant increase in IADL six... "months" should be added.

Author Response

Dear Reviewer,

thank you for your diligence. Your comments substantially improved our work ! All changes in the uploaded manuscript were highlighted in yellow color. Our responses to your comments here are highlighted in orange.

Introduction section

1.The authors focus on activities of daily living (ADL) to establish perioperative risk factors of functional decline half a year after cardiac surgery and intensive care treatment. I wonder why the focus is on ADL and not on frailty, which is often the case in other studies and could have contributed to better comparability? Because on page 2 in line 56 the authors briefly discuss frailty.

Patients scheduled for elective cardiac surgery were expected not to be frail, in contrast to patients with a fractured neck of femur or receiving a transcatheter aortic valve implantation (TAVI). Apart from this, no standard frailty assessment exits. Some tools, such as gait speed, are not easily assessed, as patients need cardiac monitoring or might be higher risk for angina. Also, we planned to have the tests repeated on telephone calls. We therefore felt the ADL reported by the spouse to be more appropriate and safe. Hand grip strength is a feature we would not like miss if we did such a study again, but this feature, again, would be reproducible via phone.

In the ‘methods’-section, it now reads (line 87 – 91):

“ACDS-ADL is often preserved at very early stages of cognitive impairment and – together with the Mini-Mental State Examination (MMSE) – may therefore be a tool to identify and monitor experiencing cognitive and functional decline over time. Also, as ADCS-ADL is completed by the spouse or close relative, it can be performed during a phone call and does not rely on the performance of the patient.”

Furthermore, it is not quite clear to me from the introduction why the period of 6 months was chosen as the observation period. Is there a reason for this? Because the authors themselves mention in their abstract (pg.1, line 27-28) that statements regarding unfavorable outcomes are difficult to make because of the different assessment periods used in other studies.

This follow-up period of 6 months was selected because most patients were expected to be back home and able to be contacted then. We expected from previous studies, that mortality is increased for up to three or four months, but not thereafter. We did not want to contact patients a year later, because we felt, that too many things may not be recalled properly, and motivation to cooperate might be lower then. So, after all, we agreed upon 180 days after surgery to be the best compromise.

The following was added to the methods section (line 130 – 133):

“The period of six months was chosen, because patients were expected to have return from hospital and rehabilitation by then, and motivation to cooperate would be higher than a year later.”

Materials and Methods section

1.From a statistical perspective, it remains unclear to me why no case number planning or power analysis was conducted for this study. Furthermore, it is a secondary analysis, for what were the corresponding data from 2009 and 2010 already used and why was a new recruitment not started again for the current study, since on the one hand the data were already used as well as are 10 years old and on the other hand it is evident in the results section (pg. 3, line 147-164) that many missing values exist in the current sample.

This study was planned as an exploratory study primarily to identify the risk factors of delirium, and secondly to investigate the factors that might have an impact on outcomes. Given this pure explanatory character, we did not attempt a power analysis, but simply planned to access all patients admitted to cardiac surgery for a half year to get the full picture of our elective patients.

2.Regarding the inclusion and exclusion criteria: Only patients over 50 years of age are included (pg. 2, line 70). However, it is not quite clear to me what the age range in this study was (age 62-77?) or on what age range the definition of elderly patients (pg. 1, line 27) in this study is based on. Furthermore, I ask myself whether all psychiatric diseases were generally excluded?

Any condition to prevent thorough cognitive assessment was excluded. This would have included psychiatric disorders, if suggested by patients’ medical records and/or ‘suspicious’ concomitant medication. But there were none.

In the ‘methods’-section, it reads (line 71)

“Exclusion criteria were (…) inability for preoperative assessment of cognitive function (aphasia, psychiatric disease… etc.).” – “Depressive symptoms were controlled for by GDS.”

Regarding age: to clarify your question, a line ‘age 50 – 59 years’ was included in Table 1. Surprisingly, though there were a few patients younger than 60, age was not a risk factor for adverse outcomes.

3.For me it is still unclear, how, by what means and who obtained the demographic and clinical data (pg. 2, line 73-74) of the respective patients? Or were they taken all from the patient's medical record?

Patients’ demographics were obtained from both medical records (e.g., medication, diagnoses), and interviews were completed after the MMSE and GDS assessment.

The following was added to the ‘methods’-section (92 – 95):

“Patients’ preoperative assessments, including documentation of co-morbidities and concomitant medication, were carried out by medical Ph.D.-students, who were repetitively re-instructed by the psychiatrist of our research team.” (J. Popp).”

Line 116 – 119: “The assessments in the ICU were done by the attending ICU physicians” (N. Theuerkauf, U. Guenther).

4.Furthermore, how long did the examination/assessment last for the patients with regard to the study relevant parameters? (pg. 2, line 73-92)

The entire assessment took about 30 to 45 min in a quiet room.

5.The authors made the following classification for nicotine consumption: "never", "stopped before at least two months", "ongoing". Why is there no "occasionally" classification, as for alcohol consumption?

It is true, “ongoing” could have been given more detailed. ‘Packyears’, however, also a widely used item, were 13 years (95%-CI, 8-17, ADL-decline) vs. 10 years (6-13, No decline), P=0.516. We did not want to give redundant information for a factor that did not reach statistical significance.

Furthermore, it is not clear to me whether the alcohol consumption is related to the type of drink (beer or wine etc.). Or has the definition of a standard drink (in the United States, one "standard" drink (or one alcoholic drink equivalent) contains roughly .6 fl oz or 14 grams of pure alcohol) been considered for this? Because it is not only the quantity alone that could make a difference.

True. Patients were indeed asked for the type of alcohol (beer, wine, shots, etc.), but here, the assessors were given some space of interpretation. Particularly the category “heavy” was judged by the assessors, never by the (few) patients. Which may have been, of course, influenced by subjective impression. To our surprise, there was no statistical correlation to outcomes.

6.The authors decided to use the ACDS-ADL for the measurement of the functional decline. It is not clear to me why the authors chose this particular assessment tool. Because the ACDS-ADL normally evaluates the competence of patients with Alzheimer's disease (AD) in basic and instrumental activities of daily life (ADLs). This study did not exclusively include patients with AD.

It is correct that the ADCS-ADL scale was designed for research in Alzheimer’s disease. The questionnaire allows for the assessment of (changes in) everyday function independently of the aetiology. It can identify functional impairment at pre-dementia stages and discriminates well between normal controls and subjects with mild cognitive impairment or dementia (Perneczky et al. 2006; Cintra et al. 2017). As no specific instrument was validated for functional decline after surgery or delirium, we used ADCS-ADL as a well-established and widely used tool to measure ADL and their changes over time.

The following was included in the methods section (line 87 – 91):

“ACDS-ADL is often preserved at very early stages of cognitive impairment and – together with the Mini-Mental State Examination (MMSE) – may therefore be a tool to identify and monitor experiencing cognitive and functional decline over time. Also, as ADCS-ADL is completed by the spouse or close relative, it can be performed during a phone call and does not rely on the performance of the patient.”

7.The decision for ACDS-ADL is referred to in one sentence (pg. 2, line 85-87), but a more detailed explanation or specification of the quality criteria of the assessment tool would be helpful.

We have now added a more detailed specification on this tool, see answer to point 6.

8.The authors make a classification of the patients in who had or did not have functional decline based on the ADCS-ADL (pg.2, line 91-92). This raises the question whether there was a corresponding cut-off for this?

We tried to clarify this. In the ‘Methods’ section, it now reads (line 134 – 136):

“Patients demographic data were grouped into whether they had a postoperative ADL lower than their preoperative ADL (‘ADL decline’), or whether there was no change or a higher ADL than before surgery (‘No decline’).”

9.After 6 months, the authors contacted the patients' relatives to perform the telephone version of the MMSE (pg. 3, line 127-129). How it was handled, since the maximum score of the telephone version was 22 points compared to the face to face version, where the maximum score was 30 points? Furthermore, how it was dealt with when no relatives could be reached? A more detailed explanation would be useful here.

The fact that we used a telephone version of MMSE is certainly a limitation. We made this choice not only because of much higher costs and efforts needed to perform MMSE as before surgery, but also because this would have resulted in a more important drop-out rate at 180 days, likely related to higher impairment in cognition and ADL.

The following is now included in the ‘limitations’-section (line 326 – 335):

„Fourth, the postoperative assessment of the MMSE was performed by a telephone version with a maximum of only 22 points. It skips those tasks that require a face-to-face situation, such as drawing the interlocking pentagons, etc. We found a significantly lower telephone-MMSE in ‘ADL-decline’-cohort. Though the 22-item telephone-MMSE was shown to strongly correlate with the full MMSE (34), we did not attempt to compare it with the pre-operative MMSE, since we felt that a telephone-MMSE at home half a year later should not be compared to a face-to-face situation in-hospital. For instance, diminished hearing was shown to be associated with lower scores on the telephone version (25). As this was the case for all participants, the lower telephone-MMSE in the ‘ADL-decline’-group might anyway be taken as a hint for a cognitive decline, but no further conclusion should be drawn.“

Results section

1.When the inclusion and exclusion criteria were first mentioned (pg. 2, line 70-72), the authors did not mention that participation in another study was an exclusion criterion, but they did mention it here (pg. 4, line 148). Please clarify.

“Participation in another study” was added to the ‘methods’-section (line 73).

2.The authors were able to determine from their data that more men than women were lost in follow-up (pg. 4, line 162). Were there reasons for this? This would be interesting to know for the conduct of future studies, so that this circumstance could be counteracted in advance.

Well, sorry, it reads: “In brief, more woman than men were lost to follow-up (…)”. We looked through our notes over and over again. My very personal impression is (UG), that men more often were unwilling to give report on their wives (I suspect bad outcomes), and women were more often reluctant to decline. Because those, who were not willing to complete the questionnaire, but stated that patients were “much worse than before”, “only lying on the couch because of back pain since the operation” etc. were exclusively women.

We can reliably exclude that they were dead. Much more patients and relatives than we expected refused to participate on the phone, but we did not urge too intensely, since anybody may withdraw consent without giving a reason.

3.For the formatting of Table 1 (pg. 4, line 174 as well as pg. 5) and Figure 1 a) and b) (pg. 6, line 200-208) see comment "Finally".

This was corrected. Thank you.

Discussion section

1.The authors should, as usual in a discussion, briefly mention in a few sentences the aim of the study before summarizing the results and discuss these (pg. 6, line 221).

The following was added to the beginning of the ‘discussion’ (line 247):

“This study was conducted to investigate the functional and cognitive status before and after cardiac surgery, and to explore potentially modifiable risk factors of adverse outcomes.”

2.In the section limitations on page 8, line 299-306, the authors have inserted the exact same text passage as line 293-299 a second time. In addition, an error has crept into this repeated text passage in line 300 after reference 33, in the way that the journal name and the year appear in the text.

Sorry – this is a strange mistake. This was corrected. Thank you !

Conclusion section

1.The authors conclude that additional work should be directed towards tool development for reliable detection of risk factors for post-operative functional decline (pg. 8, line 311-313). This is important in view of the large number of different assessment tools. It would be nice to add recommendations or ideas for the development of such a reliable tool, which have the authors drawn from their own experience or could they suggest.

This was done. What we have learnt is, that (1) one cannot draw a conclusion from functional status to cognition, but (2) cognition is key, not functional status. (3) QoL may be independent from function, but - as it seems – not cognition. So, for quality control, we need these three items be measured by simple tools, and be able to reproduce them on the phone after surgery.

The ‘conclusion’ now reads: (line 349 - 350):

“Hence, additional work should be directed towards tool development for reliable detection of risk factors for post-operative functional decline. Ideally, these should include brief measures of cognition, functional status and QoL, and it should be possible to perform them during follow-ups on the telephone.”

Miscellaneous

1.Finally, the article is inconsistently formatted. The formatting would have to be revised again to make everything consistent. E.g. Introduction (on line 43) should be the headline on the page where the corresponding text passage is located (the same on line 146 concerning results).

This was corrected.

2.Regarding table 1 and figure 1: The percentage signs (%) in the table should be formatted without spaces behind the numbers and uniformly positioned one below the other (Table 1 on line 174).

This was corrected.

In figure 1, the a) and b) should be on the same level (pg. 6, line 200). In addition, on page 6 in line 207 lacks the corresponding explanation of the **.

Explanation for ** was included.

Minor points:

  • Page 3, line 136 as well as line 137: strange characters (they are not displayed correctly for Apo 4 allele and Chi. The same applies to page 7, line 226).

The strange characters were replaced. Let’s hope, that they are not change during further handling of the manuscript.

- Page 6, line 225: One blank too many before factors.

This was corrected.

- Page 8, line 296: Accordingly, even a significant increase in IADL six... "months" should be added.

Thank you, “months” was added.

Again - thank you for your meticulous work !

Sincerely,

Ulf Guenther

Reviewer 3 Report

Thank you for the submission of this interesting article.

The long-term outcome of our patients is unfortunately still a very important parameter for the quality of our medical work. I therefore strongly support the consideration of ADL 6 months after surgery.
This examination should become the standard in clinical routine in the future.

Do you think, that the premedication with midazolam (7.5 mg) is the best way to treat your patients?

Can you explain, why you used MMSE instead of MOCA in your article?

Author Response

Dear Reviewer,

thank you for rewiewing our work. Your comments substantially contributed to our work. Please find below the replies to your comments, highlighted in orange. In the uploaded manuscript, changes are highlighted in yellow color.

Do you think, that the premedication with midazolam (7.5 mg) is the best way to treat your patients?

We feel, that more often than not, a pharmacological premedication is not necessary, particularly in elderly patients. There are, however, data suggesting that benzodiazepines are only associated with a higher rate of delirium (1112 ICU-patients) if administered through continuous infusion (Zaal, Devlin, Hazelbag et al., ICM 2015). Based on this, premedication in our institution is left to the attending anaesthesiologists’ discretion.

Can you explain, why you used MMSE instead of MOCA in your article?

While MMSE appears to be a useful tool to screen for dementia, it has important limitations to quantify cognitive performance in subjects that may have only mild cognitive impairment or normal cognition, as in our study. MOCA seems now more appropriate to use, indeed. Unfortunately, the German version of MOCA was validated only after we performed our study.

The following was included in the ‘discussion’-section (line 258):

As a measure of cognitive performance, the MMSE was used here, since it had become a standard in this field of research. MMSE alone, as a single tool, however, is not sufficient for the diagnosis of cognitive impairment. The purpose here was to identify single factors to identify patients at risk of functional decline in the clinical setting.”

Again, thank you for your comments.

Yours sincerely,

Ulf Guenther

Round 2

Reviewer 2 Report

Thank you for the opportunity to re-review your work.

In this revision of the manuscript, all my comments and suggestions were addressed by the authors appropriately.